# A Literature Review of Big Data-Based Urban Park Research in Visitor Dimension

**Hongxu Guo †, Zhuoqiao Luo †, Mengtian Li, Shumin Kong and Haiyan Jiang ***

School of Architecture and Urban Planning, Guangdong University of Technology, 729 East Dongfeng Road, Guangzhou 510090, China; guohx@gdut.edu.cn (H.G.); 2112110019@mail2.gdut.edu.cn (Z.L.); 2112110043@mail2.gdut.edu.cn (M.L.); 2112010057@mail2.gdut.edu.cn (S.K.)
* Correspondence: jianghaiyan@gdut.edu.cn
† These authors contributed equally to this work.

**Abstract:** Urban parks provide multiple benefits to human well-being and human health. Big data provide new and powerful ways to study visitors' feelings, activities in urban parks, and the effect they themselves have on urban parks. However, the term "big data" has been defined variably, and its applications on urban parks have so far been sporadic in research. Therefore, a comprehensive review of big data-based urban park research is much needed. The review aimed to summarize the big data-based urban park research in visitor dimension by a systematic review approach in combination with bibliometric and thematic analyses. The results showed that the number of publications of related articles has been increasing exponentially in recent years. Users' days data is used most frequently in the big data-based urban park research, and the major analytical methods are of four types: sentiment analysis, statistical analysis, and spatial analysis. The major research topics of big data-based urban park research in visitor dimension include visitors' behavior, visitors' perception and visitors' effect. Big data benefits urban park research by providing low-cost, timely information, a people-oriented perspective, and fine-grained site information. However, its accuracy is insufficient because of coordinate, keyword classification and different kinds of users. To move forward, future research should integrate multiple big data sources, expand the application, such as public health and human–nature interactions, and pay more attention to the big data use for overcoming pandemic. This review can help to understand the current situation of big data-based urban park research, and provide a reference for the studies of this topic in the future.

**Keywords:** perception; social media; mobile phone data; health

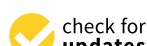



## 1. Introduction

Urban parks have generally been considered as places of welfare and public good that offer a variety of benefit to citizen, including health and happiness. Visitors' feeling, activities and effect they have on urban parks will influence the planning and management of the parks. Therefore, studies on visitors' satisfaction, well-being and emotion are hot topics [1,2]. Over the past few years, most of the studies have entailed the use of questionnaire-based surveys, onsite semi-structured individual interviews, and systematic observations of urban residents [3–5] to obtain accurate knowledge and insights regarding users' subjective evaluations and experiences. However, the application of these methods is usually site-specific and time-consuming, and the use of limited samples constrains data availability for the studies of large populations [6] and may not accurately reflect the understanding of urban agglomeration [7].

At the beginning of 2000s, people's lives have been influenced by digital data as more people started to use computer and smartphone. In this moment, big data was launched by the Special Issue in Nature in 2008 and defined as a system with huge amount of information and complex software [8]. After that, the concept of big data went vital and was widely used by researchers because big data, such as mobile phone signal data, social

media data, Open Street Map (OSM), can help to deal with problems in urban parks. With the popularization of iPhone and other smartphone installed with Android System, many social media started to apply in smartphone in these 10 years. Thus, the location-based mobile social networks can cover more information of human beings and collect spatial information and social communication of human nimbly and reliably. First, GPS and mobile base stations are used to collect mobility records, such as ID, trajectory, coordinate, and time of visitors [9]. China's scholars have used mobile phone data to examine the visitor mobility in metropolis, such as Beijing [10], Shanghai [11], and Nanjing [12]. These findings provide new insights into the trajectory of visitors at fine spatiotemporal scales, contributing a lot to local municipal policymakers in the planning of an urban park system [13,14]. Second, social media, such as major social networking sites Facebook and Twitter, have gained popularity across nations, cities, and demographic groups since the beginning of the 2000s, and there have been numerous attempts in academia to define and describe the potential roles of social media in our society [15]. Different elements of user-generated data sets, such as geotags, timestamps, content, and user information, provide new possibilities for studying the thought of visitors from different perspectives [16]. Third, mobile application, mobile signal and social media data can not only collect the reaction and mobility of visitors but can also combine with OSM, Baidu map, and Remote Sensing (RS) to assess park service range, environment services, and surrounding transport, revealing the problem happened in public health. For example, some researchers assessed the cultural ecosystem service in support for maintaining the sustainable development of urban parks for human well-being and ecological conservation of park green space. In addition, some research used OSM and questionnaires to assess whether the activities happened in parks can improve the health quality of visitors [17]. Using big data to study trajectory of visitors, thought of visitors affect human health and the understanding of human–environment relationship in parks. Additionally, it can optimize park planning, management and design to create a human-friendly urban park.

Some reviews have summarized various related research, such as analyzing big data-based urban sustainability analysis [18], research on parks and protected areas by social media [19], and an urban green space study based on VGI (Volunteered Geographical Information) and social media [20]. These studies are highly useful as they reported numerous new insights that could not obtained by traditional research approaches. Big data-based urban sustainability research includes various kinds of big data and studies diverse themes, including urban mobility, urban land use, urban planning/design, environmental sustainability, public health and safety, social equity, tourism, resources, and energy utilization, real estate and retail, accommodation, and catering, but this kind of research could not give strong advice for urban parks. The data source of other two reviews were also limited because it only focused on social media and VGI, but not included eye-tracking, video data of drone and card transaction data. The review of VGI was lack of a systematic summary of the topic, and the review of social media was uncompleted, similar to the visitors' effect. Therefore, there is still a need to compare the applications of big data in different types of study on urban parks in the visitor dimension.

The main objectives of this study were to provide a comprehensive and quantitative assessment of big data-based urban park research in visitor dimension by the systematic review method together with quantitative (bibliometric) and qualitative (thematic) analyses, so as to provide new ideas for future studies of this topic. This paper is organized into 5 sections. Section 1 is the introduction, which presents the background of big data-based urban park research in visitor dimension and our framework. Section 2 "Method" reveals the preparation and process of this review. Section 3 "Result" describes the application of big data in urban park research quantitatively and qualitatively, and analyzes the spatiotemporal characteristics, data types, analytical methods, and topics in urban park research in visitor dimension based on big data; Section 4 "Discussion" summarizes the advantages, limitation and future directions of big data for urban park research and Section 5 "Conclusions" presents the major findings and conclusions.

## 2. Method

The review approach aims to answer specific research questions by collecting all the empirical evidence from eligible studies. It uses clear, systematic methods to summarize compelling evidence, provide reliable findings, and then draw solid conclusions with the advantages of quantification via bibliometrics and flexibility after thematic analysis.

According to the steps proposed by Liberati [21] and Khan [22], literature search uses the core collection of the web of science because this collection is more reliable, and has more authoritative researchers. The review mainly focused on the big data used in the studies of the visitors in urban parks, and excluded the articles that only focused on traditional data, such as questionnaire and field survey.

The keywords used in the literature search included "urban park" and "big data OR social media OR phone OR smartphone POI OR Twitter OR Facebook OR Sina OR Wechat OR app", and the search period was from 1 January 2011 to 31 December 2021.

Five hundred and fifty-six papers were found according to this search criteria and were further filtered according to the following steps: (1) look over the title and abstract and then drop the topics that are only about driving, traffic, parking, smart city, industrial park, street food, business and retail park, market, clinic, pollution, education park, recreation, protest, and agriculture and (2) check the research method and remove the review papers just using the traditional method, just locating pre-urban, suburban, and the research not available (Figure 1).

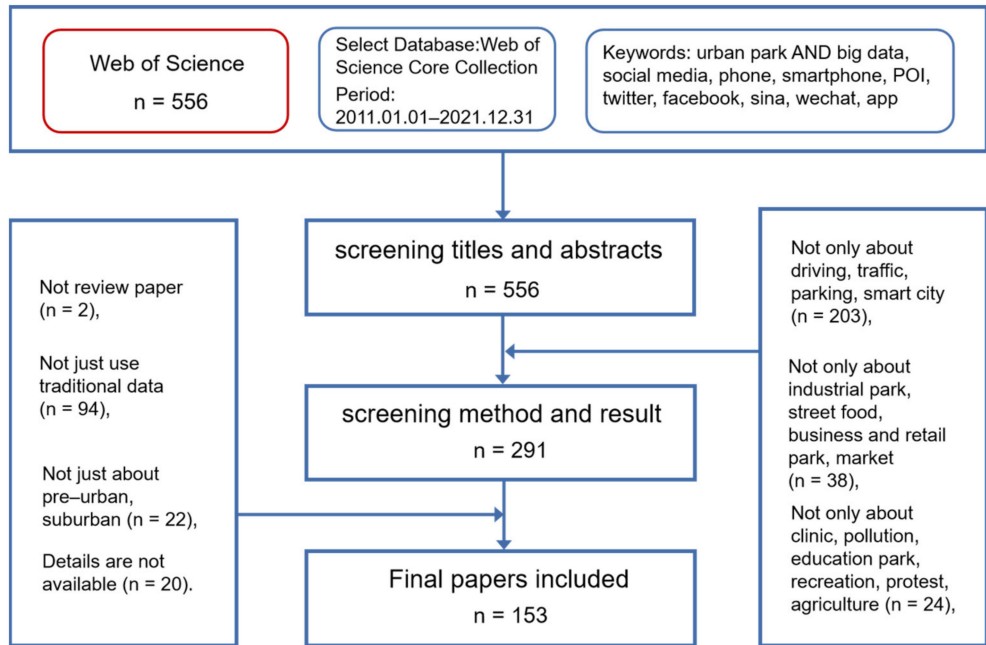

**Figure 1.** Flowchart of screening effective urban park-related literature.

Ultimately, 153 papers that met the criteria were identified. According to the keywords, titles, and abstracts, the papers were classified into 3 themes including visitors' behavior, visitors' perception, visitors' effect. Based on the selected literature, the types of big data, the methods for analyzing big data, and the key findings for each theme were summarized.

## 3. Results

### 3.1. Urban Park Research by the Numbers

The big data-based urban park research has showed up in 2011 (Figure 2) and the number of these researchers started to increase dramatically from 2017 and it showed an upward trend although there was slight fluctuation. As for the times of cited, the number of citations kept stable from 2011 to 2015. It rose rapidly after 2016 and reached 1000 in

2021. This graph shows that the aspect of big data-based urban park research become more various, and more researchers have concentrated on this topic. According to their study areas, the big data-based urban park research covered 67 countries across six continents (Figure 3). The percentage of the urban park in Asia was the largest, which occupied 37.2% (281 articles), followed by Europe (236 articles, 31.3%) and North American (169 articles, 22.4%). Figure 3 also showed that there were many authors from China, the United State, and the United Kingdom studying urban parks.

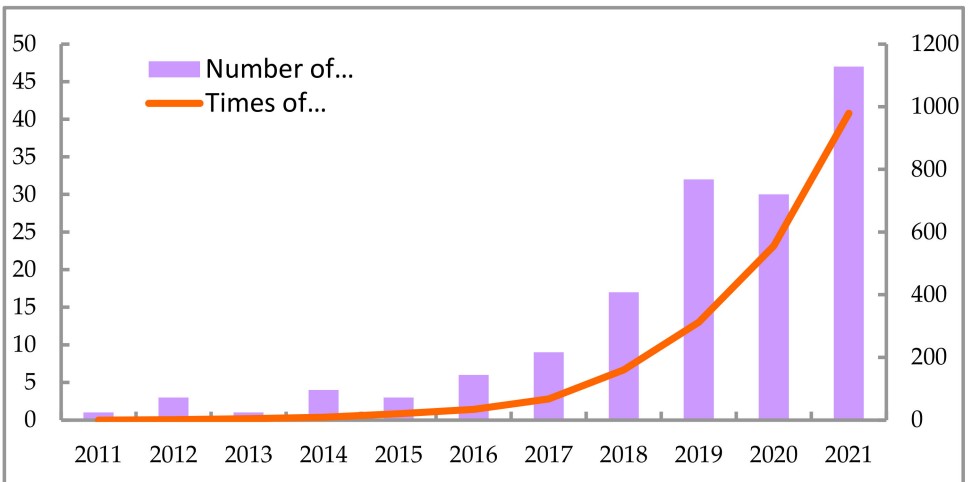

**Figure 2.** Numbers of publications and citations for urban park research using big data.

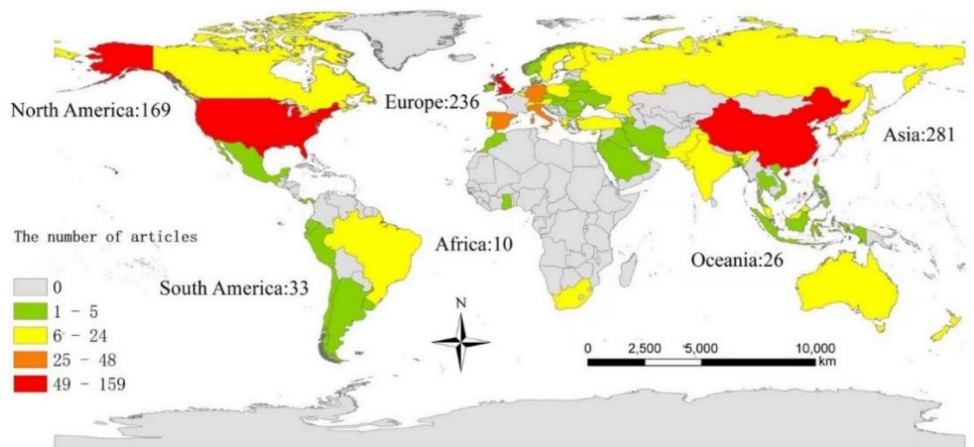

**Figure 3.** Spatial distribution of publications in urban park research using big data.

These urban park research articles were published in 80 journals, and the journals with the largest numbers of relevant articles were mainly concentrated in the fields of environmental sciences ecology, urban studies, and geography (Figure 4), including some conference papers. Among these journals, Urban Forestry & Urban Greening had the largest number of publications, accounting for 9.8% (15 articles) of the total, followed by Landscape and Urban Planning (13 articles, 8.5%) and Sustainability (13 articles, 8.5%).

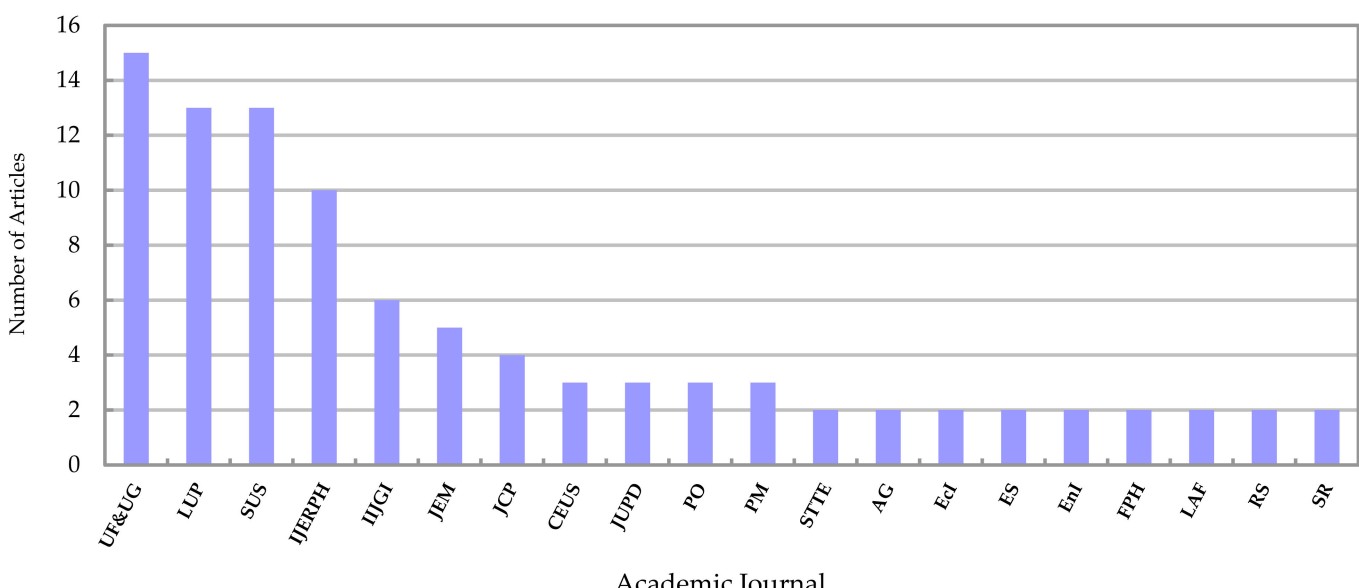

**Figure 4.** Distribution of papers in urban park research using big data indifferent journals. Note: UF&UG: Urban Forestry & Urban Greening; LUP: Landscape and Urban Planning; SUS: Sustainability; IJERPH: International Journal of Environmental Research and Public Health; IIJGI: Isprs International Journal of Geo-Information; JEM: Journal of Environmental Management; JCP: Journal of Cleaner Production; CEUS: Computers Environment and Urban Systems; JUPD: Journal of Urban Planning and Development; PO: Plos One; PM: Preventive Medicine; STTE: Science of The Total Environment; AG: Applied Geography; EcI: Ecological Indicators; ES: Ecosystem Services; EnI: Environment International; FPH: Frontiers in Public Health LAF: Landscape Architecture Frontiers; RS: Remote Sensing; SR: Scientific Reports.

### 3.2. Types of Big Data Used in Urban Park Research

Big data, which can be classified into different types according to various criteria, such as data source, data environment, and data geometry, have been used to describe the real-time status of urban environmental factors, economic activities, and the behavioral patterns of urbanites. In this study, the classification was divided into six types based on the attribute of data sources; they are user days data, comment data, map data, phone data, image data, and other data (Table 1).

User days is a data source that contain users, coordinate, date information, text or photograph within a given period [23] on Twitter, Sina, Flickr, Instagram, Jiepang and Facebook. Twitter and Flickr are used the most on urban park research, which can represent tweet and photo user days, respectively. This data can get more information than other kinds of data, but they are also limited by the age and the location of users. Comment data includes Tripadivsor, Ctrip, Dianping, and platforms having user reviews. This kind of data source always provide the review of the users who went to some places and wrote some reviews to express their feelings. The reviews contain titles, bodies, rate-values and dates [24]. It has some limitations that it cannot get the detail information of the reviewers, and the keyword of the comments may be not useful for judging the type and value of the service. In addition, it also cannot get the elderly and children comments. The map data including POI, heat map, and route map, is usually used widely in helping other kinds of data source to solve problems, including AMAP, Baidu, OSM, and Google map. These map platforms provide an objective view of services in the park, but they could not estimate the exact number of differences between different urban parks. Mobile signal data and phone application are phone data, both of which can provide the trajectories of visitors, consider the elderly can be considered and they can add new way for accessibility. However, they are limited by the location of source. For example, China Mobile company can only provide the information of their own country, even own company, and has high measurement error

level at the border area. Image data includes data source of images and videos because the video will be turned into images when being analyzed, and eye-tracking, RS and drone are belonged to image data as well. RS often combines with other kinds of big data, but eye-tracking is always used in alone. Other data include acoustic data and transaction data.

User days data were used in 50 publications (31.6% of all the relevant articles), the commonly used big data in urban park research, followed by phone data used in 34 articles (21.5% of all the relevant articles) and map data (18.4% of all the relevant articles).

*3.3. Main Methods for Analyzing Big Data in Urban Park Research*

Big data analysis refers to the process of discovering potential value from big data [25]. Different methods mine data from distinct perspectives and may produce different findings; they are often associated with certain types of big data and have specific purposes (Table 1).

These methods are mainly classified into two steps, preprocessing and processing, and each of these steps is described briefly in the following contents. The first step is pre-processing, which includes the data acquired directly from database and companies or governments information by web crawling tools. The data from the company or government database can be gained by downloading from open websites or purchasing from special websites. Lin used phone data that were acquired from Fuzhou China Mobile company to explore the disparities in park accessibility [26]. Crawling is a convenient method of downloading data from websites, including Python, JSON, Java, Gooseeker search, and API. This technique is used mainly for acquiring some textual or location information from platforms, such as user days (Weibo, Twitter, and Instagram) and comment data (Dianping, Ctrip, and Tripadvisor). Ullah used JSON files to crawl Weibo data, including a unique user ID, check-in, date, time, geographic location (latitude and longitude), and gender. After gaining information by crawling software, data may need data cleaning, data transformation and so on [27].

Processing is the second step with sentiment analysis, statistical analysis, and spatial analysis. Sentiment analysis is used in user days, comment data and image data to analyze the frequency of word use, sentiment of sentences and the sentiment in pictures. There are mainly five methods: Amazon's Mechanical Turk, Latent Dirichlet Allocation (LDA), word2vec model, ROST Content Mining 6 (ROST CM6), and FireFACE V1.0 software (Zhilunpudao Agricultural Science & Technique Inc., Changchun, China). Amazon's Mechanical Turk posts the task on the website and word ratings are calculated by averaging scores from a pool of online crowd-sourced workers [28]. LDA is a kind of unsupervised learning that can find the hidden theme of the sentences and cluster similar words [24]. FireFACE V1.0 software is a kind of image analysis software trained by 30,000 photos to recognize the emotion of visitors [29]. Statistical analysis is used the most, among which SPSS, R, Tableau, and two-step floating catchment area (2SFCA) are used to realize the relation between visitation and its factors. All kinds of data can use statistical analysis, especially in the Student's *t*-test, Analysis of Variance (ANOVA), logistic regression model, Mann–Whitney test, Spearman's correlation, and Pearson correlation. *t*-test is an analysis method that can make comparison between two situations [30], and ANOVA is used to show the connection between various factors and main elements [31]. Spatial analysis is a part of Statistical analysis, and Geographic Information System (GIS) is the Spatial analysis that is used in almost all kinds of platforms, and its Global Moran's I, Kernel density estimation (KDE), and geographically weighted regression (GWR) are used widely in nearly all kinds of data. Moran's I can be seen in many studies as it can classify different areas and analyze the connection between each area [32].

**Table 1.** The major characteristics (pros and cons) and analytical methods of big data in urban park.

| Type | Data Source | Characteristic | Advantage | Disadvantage | Data PreProcessing Method | Data Processing Method |
|---|---|---|---|---|---|---|
| User days data | Twitter | Twitter allows user to use brief words, sentences, emoji, videos and photos to share feelings in daily life [33]. The location and tag can be gained [34]. | Research can access full information of visitors in their timeframe [35]; user days data can discover new insight in public investigate [34]; many people choose user days data as it owns a huge volume of global data [33]. | Flickr data is limited by forested urban parks while Twitter is more diverse [33]; users in user days data are only the wealth young [34]; user days data will be out of data when users do not tweet or upload photos [33]; the locations user days data provide are not accurate enough [36]. | Python [30], JSON [37], GooSeeker [38], Java [39] | Amazon's Mechanical Turk [28], NRC Emolex [40], Moran's I [41], *t*-test [30], logistic regression model [42] |
| | Flickr | User in Flickr can post their photo and their location [23]. | | | | |
| Comment data | Tripadvisor | Tripadvisor has user reviews which researchers can collect the titles, bodies, rate-values and dates [24]. | The satisfaction of park users can be easily accessed and the reason of their favor can be known [43]. | It is difficult to access the gender, age, occupation and income of users; the keywords in the comments may not convenient for assessing cultural service's type and value in park; only get comments from young people [44]. | Python [43], GooSeeker [44] | LDA [24], ROST CM [44], word2vec model [43], multiple regression model [45] |
| | Dianping | Dianping is a popular social media platform in China that can provide the image and textual review [43]. | | | | |
| Map data | Baidu | Baidu Map can be divided into two types, Baidu heat map and POI. Baidu heat map can show the density and flow of people [46]. Generally, POI data contains spatial features of name, category, longitude, latitude, etc. [47]. | Quantitative evaluation provides a relatively objective and visual way to understand park services [32]; it was better than the data obtained from Weibo check-in data in reflecting the real conditions [46]. | It could not estimate the exact number of differences between different urban parks [46]. | API [48] | KDE [49], GWR [50], network analysis and buffer analysis [46] |
| | OSM | Route data and public transportation is gotten from OSM [32]. It often combines with other kinds of data. | | | | |

**Table 1.** *Cont.*

| Type | Data Source | Characteristic | Advantage | Disadvantage | Data PreProcessing Method | Data Processing Method |
|---|---|---|---|---|---|---|
| Phone data | Mobile signal | Mobile signal data contains the ID, trajectory, coordinate and time of user that are always gained from companies in different countries [45]. | Mobile signal data can be used to detect the behavior of the elderly; accessibility can be assessed in a new way [51]; Multiple mode of actual travel behavior can be obtained by alleviating the limitations on the lack of human data [10] | The information is limited by the location [52]; they have high measurement error level at border areas [52] | | Gaussian-based 2SFCA [26], GWR [26], *t*-test [53], Mann-Whitney U-test [53] |
| | Phone application | Tencent, MapMyFitness, Stava and Wikiloc are phone applications. These kinds of app often provides the route of users [52,54]. | | | | |
| Image data | RS | It shows as an image that is often token by satellite [55] and often combines with other kinds of data. | Easy to operate, as it only needs photo to analyze. | Their consideration is not comprehensive. | | ANOVA [56,57], FireFACE software [29], zonal analysis [47] |
| | Video | Video data record the movement and facial expression of visitors, and what visitors see. Eye-tracking imitates the eye-movements by using video to record what people see [58]. | | | | |
| Other data | Transaction | This kind of data collect the transaction of visitors. It would collect sales information based on gender, age, time, day, and business types [59]. | It is a new way to judge visitors' effect. These kinds of data is used in interdisciplinary research, such as collecting the sound of animal and analyzing the interaction between human and animal. | They are not easy to use, especially acoustic. The data is lack of samples, and they are limited by the high cost. | | Mann–Whitney test [60], Zonal Statistics tool [61] |
| | Acoustic | It collects the sound inside parks and finds whether the noise of visitors would influence the nature [60]. | | | | |

### 3.4. Key Themes in Big Data-Based Urban Park Research in Visitor Dimension

Visitors in urban parks can be accessed by big data in various ways, and the topics of this research can be divided into three themes: visitors' behavior, visitors' perception, and visitors' effect. These themes were mainly sorted by the titles and keywords of the research. For example, mobility and activities of visitors should be contained in behavior. Emotion, satisfaction, and well-being of visitors should be contained in perception. Effect could contain the healthy and environmental effect of visitors. Besides, each theme would mainly correspond to different sources of big data (Figure 5), and each data also has connection with different topics under the themes.

More details for each theme are as follows:

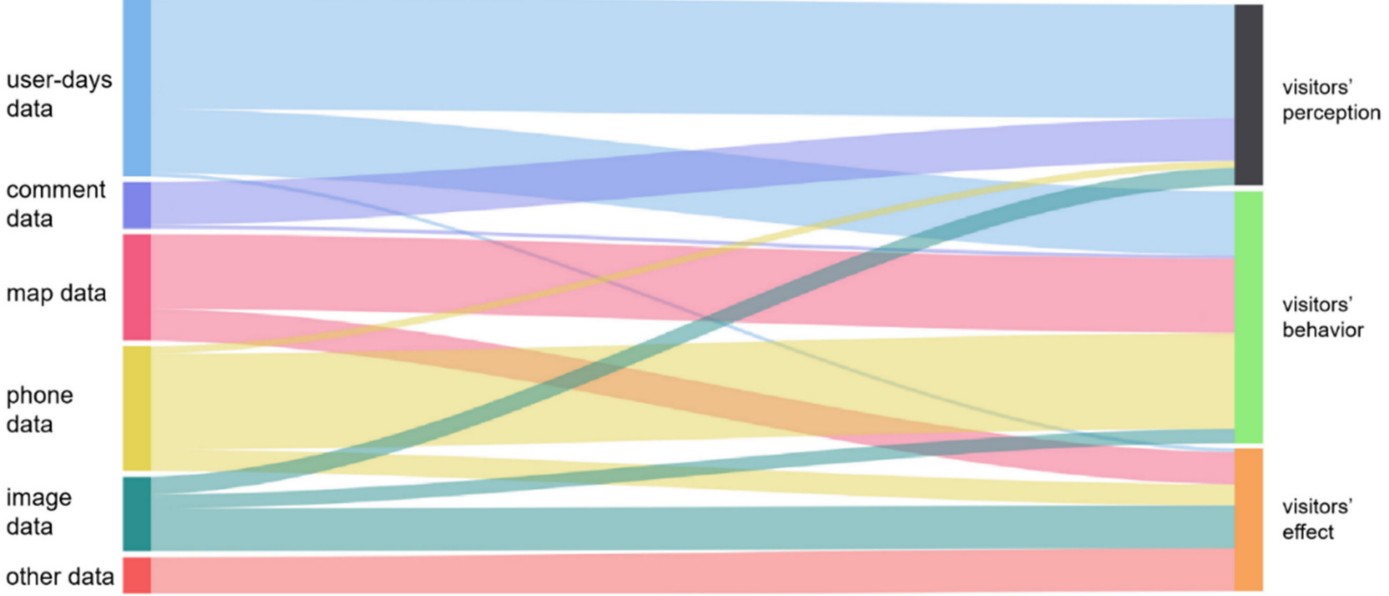

**Figure 5.** A sankey diagram illustrating the relations among different types of big data and research themes.

3.4.1. Visitors' Behavior

The spatial and temporal distribution of visiting and the activities happened in urban parks can be detected from visitors' behavior, so this theme has more chance to use big data. Phone data, map data, and some social media are the ways to collect the information.

Phone data are utilized the most in this theme, as it can study the accessibility of neighborhood, frequency of visitation and activities token part in the parks. Since mobile signal data has the coordinates, characteristics, and timestamp of visitors, researchers would more like to study the accessibility of disparities. Yang et. learned that the parks in Shanghai protected the equality of the disparities. They especially pointed out that the parks were the most inequality at workdays, and people living in different type of housing would affect the equality as well [11]. Additionally, some research collected mobile signal data to find the different frequency in different seasons on three kinds of park visiting. They found travel distant can influence visitors' frequency [62]. In addition, the activities token part in parks can combine app data and GPS data, so some researchers collected the tracking data in GPS and the activities data in the app data. They concluded the choice of visitors who do sport in parks, which may give suggestions for management of the recreational area in the park [63]. Besides, it also had research that only used app to discover park activities. They collected some sport routes data and found the distance between parks and home could influence the number of visitors [54].

The map data is mainly used to analyze the influencing factors of the visitation in the park and the reason for the number's change every week. The research used AMAP and weibo check-in data to study 13,759 parks in China and understand the factors of visitation.

After considering park attributes, accessibility, and the socioeconomic environment, they found that surrounding environment had significant positive influence while park attribute had negative influence [64]. Besides, Baidu heat map combined mobile phone data to discover the difference of visitation in weekday and workday. The result showed that there were differences in peak time and frequency of visitors at weekend and workday, and visitors occupied the area in various kinds of parks from time to time [50].

In this theme, social media were used to count the number of visitation and the seasonal various of visitation. For example, 340,000 check-in data in Sina micro-blog between 2012 and 2014 were used to count the volume of visitation to find out the park popularity and the future tendency [65]. Twitter was used to figure out seasonal differences in physical activity, and the result showed that outdoor fitness decreased from winter to summer, cycling and water sport kept stable, and team sport and fun sport increase from winter to summer [39].

3.4.2. Visitors' Perception

The perception of visitors means the physical and mental feeling of visitors, such as satisfaction, well-being and emotion. There are kinds of social media that can present the objective feelings of visitors, such as Twitter, Facebook, Sina Weibo, and some travel applications. In addition, many researchers started to turn their textual study into visitors' facial and visional study. These kinds of studies can enrich the range of perception and use diverse dimensions to dig out the objective suggestion of visitors.

There are mainly three kinds of big data that are used to collect the perception of visitors.

Firstly, the textual data and photo data in user days data, such as Twitter, Sina Weibo, and Flickr, are used the most. They always solve the problem in visitors' happiness, the reason of visitation, and evaluate park services. The geo-located user days data and check-in data in user days data are always used to collect the visitation reasons. In addition, policy makers can also use social media to know the new thought of visitors during COVID-19. Visitors' happiness can be measured by the concept of tweets. Some study used Twitter to judge the people's emotion outside and inside the park and found that people had more positive emotion in the park in general [36]. To discover the factor of visitation, some researchers collected photos in Flickr and used Google Cloud Vision to analyze them [66]. The research found that people would be more likely to take natural photos, summarizing the value of the cultural ecosystem service in combination with the spatial distribution [66]. The researchers from the USA studied the city parks in New York by geo-located user days data. They collected the coordinate of visitors and the specific time of visiting from 2005 to 2014 and finally found that the visitation was positively correlated with transportation, athletic facilities, impervious surface, water bodies, but was negative with green space and minority in neighborhoods [34]. Comparing the feelings before and after the pandemic through social media, researchers can know the demand for the great changes in society. The researcher looked up Instagram in Hong Kong, Singapore, Tokyo, and Seoul and discovered that people would like to visit nature parks that are large and close to city centers [42].

Secondly, comment data, such as Ctrip and Tripadvisor, are used to collect comments of visitors to judge their satisfaction for assessing services and landscape in the park. Some research assessed the cultural service by comments in Ctrip by analyzing 19 urban parks in Xuzhou City, China, and figured out the classification of cultural services and the basic analysis of the perception of cultural services [44]. The study used Tripadvisor to analyze the reviews from 2010 to 2018 in Bryant Park, and the result included the collection of topics and frequency of the topic in Bryant Park [67].

Finally, eye-tracking data is new data for studying the perception of urban parks and often deals with the assessment of landscape in the park. The study combined POI and AOI to find out the reasons for how long people stay in the park and assessed the usage of park landscape by analyzing the videos that people shot when they walked in the park [58].

3.4.3. Visitors' Effect

When people visit parks, they may make some effect or receive some benefit on society, economy and environment. There are negative and positive effect which visitors may cause, and elements of the effects are diverse. Positive effect includes enhancing the health and education of visitors, and promoting the economy of surrounding parks. However, it also has the negative influence on plants and animals, and accidents may happen when people visit parks.

In this term, five kinds of database can be used to evaluate the effect of visitors, including map data, user days data, transaction data, acoustic data and image data. Map data is always used in judging the social benefit of visitation. The research assessed the public health condition by Open Street Map combining traditional questionnaire and found that low availability to the park restricted people to expend their energy, so it was not good for their health [17]. Besides, research used GIS to find out that trees in parks were linked to mental health of visitors, and poor-quality park environment, such as less clean sitting, may be barriers of physical health [68]. User days data is also used to measure visitors' health. Research collected the activities happened in urban parks via twitter to assess physical activities in urban parks, because in some literature about public health connects health with parks, the activities happened in the park can enhance the health condition of human [39].

On the other hand, transaction data are used to realize the economic effect of visiting parks. The researchers studied whether visitation of the park would influence economic situation in Korea. The results showed that the economically distressed neighborhoods could get positive effect from the visitation of urban parks, and the surrounding business were influenced by park visitation [59]. Some researchers used acoustic data to test the tolerance of animals, bird and bat in the studies. For instance, 91 bird species in 27 parks in Spain and Portugal were studied, and the researchers found that visitors should keep their voice lower than 50 dB for endangered birds living [60]. In addition, accidents may happen when people visit parks. For example, visitors may get hurt when trees fall down. The Parrot AR.Drone 2.0, which is a light-weight unmanned aerial vehicles, are used in this theme as image data to assess the plant in the park. The tree hazard rating could be assessed by AR.Drone based on six variables: Trunk Condition, Growth, Crown Structure, Insect and Disease, Crown Development, and Life Expectancy [69].

## 4. Discussion

### 4.1. Advantages

Big data has three advantages to study the visitors in urban parks—that is, low cost and time saving—new "people-oriented" perspective and flexible to gain specific site information gaining.

Firstly, recent studies have demonstrated that geospatial data describe the behavior of urban residents both at urban and regional scales featuring low cost, time-saving, and convenient [31,70]. Big data can not only capture a large volume of information quickly but also can get past and real-time information easily. Zoé A. Hamstead explored variation across New York City's 2143 diverse parks and model visitation based on the spatially-explicit park characteristics and facilities, neighborhood-level accessibility features and neighborhood-level demographics by Flickr and Twitter data that contained over 250 million publicly geotagged photographs taken between 2005 and 2014 and 51.3 million geotagged tweets posted between 2012 and 2014 [34].

Secondly, with the increasing pervasiveness of new technologies, individual-level and activity-travel data from sources such as mobile phones and location-based social media (Facebook, Twitter, etc.) have become available, which may provide new opportunities to move beyond place-based and/or infrastructure-based accessibility [71,72], enabling detailed investigation of people's physical usage of parks. Furthermore, there are the capacity and great promise of big data in assessing landscapes as there are large volumes of data available online which implicitly demonstrate users' attitudes and emotions, and

big data users post text to express their thoughts directly [73]. Big data will help park management to become more people-oriented.

Finally, these data provide new opportunities for understanding how cities function in space and time [74] and how human–nature interactions occur in different environment [60]. They can reveal the connection between the attribute of parks and visitors. Tradition methods can be used for field surveys, but big data achieved from governments or companies allow researchers to get more specific and comprehensive information. For example, based on land use data from the Chinese Academy of Sciences, patches in each urban park in the study were identified [75]. Additionally, some studies got Land cover information from the Geospatial Data Cloud site, and climate information can also easily get by using big data. Compared with a signal park or a specific region. These kinds of data can be applied in large-scale studies and will give more valuable suggestions on relevant theories and applications [76].

### 4.2. Limitations

Although big data is convenient for studying urban park research, it also has limitations, especially in accuracy.

Firstly, coordinate is not accuracy enough. Phone data and user days data have the geo-location, but coordinate only has grid. For example, in the research that used mobile application data to measure visitors' behavior, the size of a pixel is 25 m × 25 m. If the area of a park is less than 0.4 ha (covering six pixels), this kind of data source cannot be used [52].

Secondly, the keywords captured from comments are also not accuracy enough, and some keywords classification cannot judge the type and value of a park's cultural services [44]. For instance, people wrote their comments based on feelings, such as "leisure", "walking", "entertainment"; "Swimming", "running, "mountain climbing", "exercise"; "Culture", "history", and "relics", but it cannot assess the type and value precisely.

Last, but not least, big data has various types, and each type has different kinds of users. For instance, the users in user-day data and comment data are wealthy young people, so they have the ability to operate computer-things, while mobile signal data will have more users, such as the elderly. However, the information of children cannot be gained in any type of big data. When the users in platforms are different, the results are also different [36], which will cause the differences between various database, and the result may confuse. If research uses one kind of data source, or analyzes different kinds of data source separately, the result would not be universal.

In total, universality and regularity in visiting a park cannot be gained, because of inaccuracy. Some papers talked about urban parks in New York, Beijing, Tokyo, Birmingham, Melbourne, and Ancona, but these are not enough. Research should focus more on regular factors to search the way for urban parks in different countries, cultures, levels of development, and climate. These factors may include underlying surface, fee management, planning standards.

### 4.3. Future Directions

4.3.1. Integrating Different Types of Big Data and Traditional Data

Using one type of big data is still not comprehensive, so it still needs to cooperate with other types of big data or traditional data to improve accuracy. The information of visitors can also be more reasonable by combining traditional data that can survey all-aged people in parks and big data that show the features of different kinds of users.

Social media data might be attached to exact coordinates, or is more coarse POI based on place names [75]. Sports tracking data is originally captured as exact GPS points, but it is often delivered in aggregate format. Mobile phones can also be allocated to locations depending on the antenna network [77]. PPGIS data often represents markers that users have placed on a map in a web browser, and the preciseness can depend on the zoom-level and local knowledge of users [78]. Combining these types of data can upgrade spatial

accuracy, making the result more credible from the perspective of understanding where, when, why, and in what way people use green spaces and who these people are [79].

Data derived from social media platforms are primarily used by younger generations. The narrowness of the population sample leads to a major limitation in social media data collection so it should be considered by combining the traditional methods of investigation (similar to using questionnaires) and big data methods to ensure the balance of sampling [30]. Both social media data and surveys are suitable and nonreplaceable data sources that can offer different user perspectives and provide complementary information. For example, phone data and user days data can cover the elderly and the young, and phone data in different companies can provide different kinds of visitors.

Additionally, because of the difference of suppliers and topics, mobile signal data, and social media data are acquired separately. In fact, smartphone can contain not only the information which gained from mobile station but also the various information in social media. If these two kinds of data can be combined together, researchers can understand comments and emotion in social media deeply in the vision of spatial movement.

4.3.2. Extending the Application Domain of Big Data

Nowadays, people's increasing demands have made the application of big data more diversified. There are mainly two dimensions that can extend into: society and environment.

The aspect of society mainly extends the studies of visitors in urban parks, including low-income people, ethnic minority people, sociodemographic characteristics (different in gender, age (mainly children and elderly), or immigration status) and differences between tourists and residents. Different kinds of people may react differently, and the demands of children may be lower stairs or some grassland for playing. However, the elderly may need some land to walk and spaces to socialize. For instance, Sihui Guo used mobile phone data to study the accessibility of urban parks for elderly residents [10]. Xiao Ping Song used social media to find the differences between tourists and residents [80]. By using different kinds of big data and traditional method to complete the information in different ages and different occupation, the results will be more reliable for solving the public health problem.

Not only for visitors, big data also need to be used for the areas inside and surrounding the park. Urban parks are different in location, natural condition, and facilities, so various kinds of park design, planning, and management need to be different. Big data create a convenient way for the study, design, planning, and management of urban parks, and its application can extend to the fields below: (1) Choose vegetation. How to plant different kinds of vegetation inside and surround the parks in the urban core and make air condition clearer in the parks. Visitors can also be protected from the mosquito. (2) Create an aromatic environment. It can find out what kinds of aroma that can make people feel relaxed and stay longer in the park, and how to place them. (3) Improve biodiversity. Some urban parks have well nature conditions, so some small animals and vegetation in the park should be protected and how to balance nature and humans should be considered. (4) Find cultural symbols. It is necessary to find symbols that can stimulate the sense of belonging for residents and attract tourists, and apply them in the park management and design, such as activities and decoration. (5) Discover the significance of patches. Patches, such as vegetation and water, may have a positive impact on visitors' perception and behavior. (6) Manage the connection between parks. Some studies used big data to assess the landscape, which many have some tips for managing parks at a large scale in the future [81]. (7) Arrange the parks and other area surrounding the parks, such as business and an office. For people's health, using big data for parks in urban areas should not only pay attention to the residential areas but also the working places. Research can focus on it and find out how to plan squares, facilities, to improve the visitation and service levels in the parks. For example, as for biodiversity, acoustic data can assess the tolerance of bird, and survey the tolerance of human.

### 4.3.3. Facing the Emergency of Pandemic

The COVID-19 pandemic has an impact on the shape and use of public spaces and will presumably leave a trace on how we approach urban planning, design, and management in the future [82]. Due to lockdown policies, working from home makes urban parks more important. Prolonged electronic device use may increase the risk of depression and feelings of loneliness [42]. Visiting green spaces can provide a break from electronic device use [83].

As the pandemic has become normal, although home isolation reduces the chance of people contracting the new crown pneumonia and effectively prevents the spread of the epidemic, COVID-19 may still have an emotional impact on people, and people's enthusiasm for outdoor sports remains undiminished during the pandemic [84]. Furthermore, studies have shown that green spaces have a positive effect on relieving mental stress.

The normalization of the COVID-19 pandemic has led to a shift in the research in urban parks and scholars have searched for new ways to confront the pandemic. To handle this situation, more urban park research should focus on social distancing [85], demand for green quality [30], and other more specific ideas for building inside and outside environments of urban parks, such as transportation, the radius of park service areas, park-related facilities and so on. To face the situations that are similar as COVID-19 or SARS, researchers should find the universality and regularity of the pandemic situation and try to provide suitable strategies for these emergencies.

## 5. Conclusions

Big data-based urban park research has been growing rapidly and has become more important in urban lives. As for the number of publications, Asian and European scholars have contributed the most, and most of the related articles mainly focused on the fields of interaction between the human and environment in urban parks.

The resources of big data were classified into six categories, which are user days data, comment data, map data, phone data, image data, and other data. When research analyses big data, there are two steps: preprocessing and processing. The first step is preprocessing, including acquiring data directly from a company or government database and crawling data with special tools. Processing has a sentiment analysis, statistical analysis, and spatial analysis. Research on the urban park is classified into three themes: visitors' behavior, visitors' perception, and visitors' effect.

Big data are low-cost time saving, "people-oriented", and able to get specific site information but are limited by accuracy. In the future, research should integrate different types of big data with traditional data to be more comprehensive. Additionally, it is important to extend the applications of big data, such as studying different kinds of visitors in urban parks and understanding the interaction between human and nature, to fill up big-data based urban parks research and create more reasonable urban parks. More importantly, because of the normalization of pandemic, research should think more about the demand and transformation of citizens and even tourists.

**Author Contributions:** Conceptualization, H.G., Z.L., S.K. and H.J.; methodology, Z.L.; software, Z.L.; validation, H.G. and Z.L.; formal analysis, Z.L. and M.L.; data curation, Z.L.; writing—original draft preparation, H.G. and Z.L.; writing—review and editing, Z.L.; visualization, M.L.; supervision, H.G.; project administration, H.G.; and funding acquisition, H.G. All authors have read and agreed to the published version of the manuscript.

**Funding:** This research was funded by the National Natural Science Foundation of China, grant number 41501184; and the Science and technology projects of Zhejiang Province, grant number 2022C0316.

**Institutional Review Board Statement:** Not applicable.

**Informed Consent Statement:** Informed consent was obtained from all subjects involved in the study.

**Data Availability Statement:** The articles were obtained from Web of Science (https://www.webofscience.com/, accessed on 4 May 2022).

**Conflicts of Interest:** The authors declare no conflict of interest.

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
