# Peer review of "A Literature Review of Big Data-Based Urban Park Research in Visitor Dimension"

_land, doi:10.3390/land11060864_

Round 1
Reviewer 1 Report
The paper is suitable for the journal and the special issue. The topic is important and emerging. Papers on this topic have high citations and it is good that a niche - that on big data, suitable for the special issue - has been found.
The abstract summarises well the paper. However, the fact that it is a review paper should be more clearly stated also here.
The references are numerous as it should be for a review paper and relevant.
The design of the research and the discussion are good. Also, it is good that the covid pandemic relationship has been included as it is relevant for the topic of well being and public health. More could be included on this topic also on mental health and relationship to the future office.
The conclusions are soundly derived from the paper.
Author Response
Thank you for your advice. Please see the attachment.

Reviewer 2 Report
Dear all,
The issue related to Big Data management is increasing every day. It shows the relevance of conducting studies on this particular topic and from a different perspective, as is the case with applying the topic research of Big Data to Urban Park Research.
Regarding the work itself, generally, it has scientific soundness, and it is not very extensive, which I appreciate. The problems is clearly identified, the methods are well explained and the results, discussion, and conclusions are well developed and sustained by the literature. The same occurs with the study limitations and prospective research lines. In this regard, I only recommend the authors consider the following references about the different uses of big data (in the case of insular/ultra-peripheral territories and in urban green spaces as a whole) through the following references:
-Castanho, R.A., Naranjo Gómez, J.M., Vulevic, A., and Couto, G (2021). The Land-Use Change Dynamics Based on the CORINE Data in the Period 1990–2018 in the European Archipelagos of the Macaronesia Region: Azores, Canary Islands and Madeira. ISPRS Int. J. Geo-Inf. 2021, 10, 342. https://doi.org/10.3390/ijgi10050342 •
-Gómez, J. M. N. , Velarde, J. G. , Gallardo, J. M. , Almonte, J. M. J. , Aliseda, J. M. , & Fernández, J. C. (2021). The Most Meridional Border in Europe. Demographic and Environmental Changes. Peripheral Territories, Tourism, and Regional Development. IntechOpen. https://doi.org/10.5772/intechopen.97566
best,
Author Response

(The authors gave the same response as above.)

Reviewer 3 Report
this paper has some potential but there are several aspects which would help to improve it. firstly, strictly speaking it is not a systematic review - there are very specific and are about evaluating scientific evidence presented in researchb papers. You are not doing this - you are summarising trends in papers in a broad way. So please be careful and give it a correct title - probably just a "literautre review". Given that you are looking at trends in how big data have been used over time it would really help if oyu provided an overview of when the various platforms and social media were founded, emerged, became widespread etc - because the research using big data is of course reliant on the availability of data which was hardly present in 2006 the year Facebook launched or later when smart phones became more widespread. Without this the context is weak and the patterns revealed do not really tell us much. I am also suprised that you did not find papers where photographs are analysed for their content and the different means of doing this. I did not see mention of GoogleCloud, Clarify, Amazon Rekognition etc as methods being applied in research into images.
Methodologically it is weak in the description of how the searches were undertaken and sorted - only a very short section which is crucial for ensuring confidence in the results.
The catagorisation of the data and how it is used is useful. However, you do not actually define what is or is not big data in the introdcution - this is necessary in order to define what you include or exclude in the review and for all the analysis and drawing conclusions later on.
The last sentence in the conclusions is not one that you can draw from the results of the review.
The paper is full of grammatical errors and needs a complete language editing.
Author Response

(The authors gave the same response as above.)

Round 2
Reviewer 3 Report
The points I raised have generally been addressed. evn if you don't have specific information on the rise of social media and availability of smartphones you can talk a bit about this in the introduction and maybe find some data about the penetration of smartphones and social media users at the start of your review in 2012 - I am sure this has a bearing on the results, even of there is a time-lag. You could even be a bit speculative about this. The language still needs some checking in places.
Author Response
Thanks for your advice. Please see the attachment.
